# Towards Developing a Robust Intrusion Detection Model Using Hadoop–Spark and Data Augmentation for IoT Networks [note 1]

**DOI:** 10.3390/s22207726

**Published:** 2022-10-12

**Authors:** Ricardo Alejandro Manzano Sanchez, Marzia Zaman, Nishith Goel, Kshirasagar Naik, Rohit Joshi

**Affiliations:** 1Cistech Limited, 201-203 Colonnade Rd, Nepean, ON K2E 7K3, Canada; 2Cistel Technology Inc., 30 Concourse Gate, Nepean, ON K2E 7V7, Canada; 3Department of Electrical and Computer Engineering, University of Waterloo, 200 University Ave W, Waterloo, ON N2L 3G1, Canada

**Keywords:** IoT (internet of things) security, big data framework, imbalaced datasets, CTGAN, hadoop-spark, BoT-IoT

## Abstract

In recent years, anomaly detection and machine learning for intrusion detection systems have been used to detect anomalies on Internet of Things networks. These systems rely on machine and deep learning to improve the detection accuracy. However, the robustness of the model depends on the number of datasamples available, quality of the data, and the distribution of the data classes. In the present paper, we focused specifically on the amount of data and class imbalanced since both parameters are key in IoT due to the fact that network traffic is increasing exponentially. For this reason, we propose a framework that uses a big data methodology with Hadoop–Spark to train and test multi-class and binary classification with one-vs-rest strategy for intrusion detection using the entire BoT IoT dataset. Thus, we evaluate all the algorithms available in Hadoop–Spark in terms of accuracy and processing time. In addition, since the BoT IoT dataset used is highly imbalanced, we also improve the accuracy for detecting minority classes by generating more datasamples using a Conditional Tabular Generative Adversarial Network (CTGAN). In general, our proposed model outperforms other published models including our previous model. Using our proposed methodology, the F1-score of one of the minority class, i.e., Theft attack was improved from 42% to 99%.

## 1. Introduction

According to Cisco, fifty percent of all networked devices will be Internet of Things (IoT) devices in 2023, reaching 14.7 billion of devices [1]. As the number of devices and connections continue to grow, Cybercriminals are also looking for new ways of conducting sophisticated attacks. Gartner states that more than 25% of the total attacks in 2025 will target IoT devices [2].

IoT devices are considered low-power devices; therefore, these devices cannot have sophisticated programs running on them to detect attacks, such as, Denial of Service (DoS), Distributed Denial of Service (DDoS), Theft attacks, or Reconnaissance attacks. Researchers have shown that pre-trained machine learning models can be deployed on IoT devices that would avoid extensive resource usage necessary for model training [3,4]. These detection algorithms can detect attacks very fast.

Pre-trained models are important for detecting attacks. However, a framework is necessary for training and developing models that are robust and perform with high accuracy. Most of the researchers use anomaly detection and intrusion signature detection [5,6] in combination or separately to detect attacks. However, both strategies in combination show better results [7].

As we rely on pre-trained models to detect anomalies accurately, we need to consider many factors that can affect the training such as (i) the number of datasamples considered in the training, (ii) the number of features selected, (iii) the quality of the data, and (iv) the distribution of the classes in the dataset.

First, the number of samples is important in the training of any machine and deep learning model. If we do not have enough datasamples our model can be under-fitted. On the other hand, if a larger dataset is available, we can obtain accurate and robust models. For this reason, the purpose of our framework is to train many machine learning models with a big dataset called BoT IoT dataset. This dataset is composed of 132 million of normal and malicious datasamples. Processing that amount of information using a laptop takes a lot of time. We verified this fact comparing Pandas and Pyspark to upload all the files of the BoT-IoT dataset. Pandas took around 30 min to upload the files while a Hadoop–Spark cluster took 10 s. The laptop that we used for this experiment is an Intel Core I5 with 8GB RAM and two cores while our Hadoop–Spark cluster has 3 nodes with 2 cores and 8 GB RAM. In addition, Spark is 100 faster than Map-Reduce according to its founders [8]. Therefore, many researchers have created machine learning models for intrusion detection with the small version of the BoT IoT dataset [3,9,10,11,12]; thus, the models can not generalize well unseen data. In addition, training machine learning models using big datasets can incur huge resource expenses because it is necessary to have computers with GPUs and many cores.

Second, the number of features selected can impact the accuracy of a machine learning model since it can cause noise or produce under-fitting. In addition, using a lot of features can cause a waste of processing resources. In related research papers, the authors used feature engineering to select or create new features obtaining robust models after training. We adopt a similar approach as in reference [13] to select enough features to avoid under-fitting but not selecting all the features to avoid noise.

Another important factor is the quality of the data. Since we did not create the dataset, we rely on the BoT-IoT dataset [9] to create our own models. In our previous paper [13], we compare the BoT-IoT dataset with other available IoT datasets publicly available and we conclude that this dataset contains traffic from real emulated IoT sensors deployed in Node-red which were connected with the public IoT hub, AWS. For this reason, we focused on the analysis of this dataset.

Finally, class imbalance can severely impact the detection accuracy of the minority classes. Therefore, balancing the data is crucial in designing an intrusion detection system using the BoT IoT dataset, since the dataset is extremely imbalanced considering the classes. To illustrate, the Normal to DoS datasample ratio is approximately 1:10000. In the present research, we use Conditional Tabular Generative Adversarial Network (CTGAN) [14] to generate unseen data from a minority class. For this reason, decision boundaries of machine learning algorithms changed and the accuracy of detection was significantly improved.

The present paper is an extension of reference [13]. In [13], a framework was proposed to train an anomaly detection (One-class SVM) and intrusion detection system using Random Forest algorithm with Hadoop–Spark framework. In [13], we proposed a new approach to train a one-class SVM in Hadoop–Spark. In addition, we analyzed the impact of feature selection in the Random Forest classifier. However, we did not evaluate the accuracy of other algorithms such as Decision Trees, Logistic Regression, Gradient Boosted Tree, Support Vector Machine, and Naive Bayes. Furthermore, we obtained bad results detecting minority classes such as Normal and Theft classes when using the intrusion detection system with Random Forest. This present paper solves these two concerns in the intrusion detection system. The contributions of this paper are the following:1Multi-class classification algorithms in Pyspark are limited to the usage of Random Forest, Decision Trees, Naive Bayes, and Logistic Regression. For this reason, in this paper we proposed the usage of One vs. Rest (OVR) strategy to evaluate the accuracy and performance of other algorithms available in Pyspark such as Gradient Boosted Tree and SVM Linear. We evaluate all the algorithms with the entire BoT-IoT dataset and identify which is the best algorithm in terms of accuracy and performance.2The BoT-IoT dataset is an extremely imbalanced dataset; therefore, we propose the usage of a new tabular data generator denoted as CTGAN to increase the number of datasamples of minority classes and obtained outstanding results in terms of F1-score.3We compare CTGAN oversampling method with other traditional methods such as Synthetic Minority Over-sampling (SMOTE) and Adaptive Synthetic oversample (ADASYN) demonstrating its accuracy to generate datasamples.

The rest of the paper is organized as follows. Section 2 summarizes previously reported work. Section 3 explains the two proposed methodologies for multi-class classification available in Spark and data oversampling using CTGAN. Section 4 presents the experiments and the obtained results. Section 5 describes the conclusions and future work.

## 2. Related Work

This section is divided into three parts. The first part takes into account research papers that use other multi-class classification algorithms to detect attacks using the short version of the BoT-IoT dataset. The second part explains research papers that use sampling methods to reduce class imbalance problem. In the third part, we present some papers that use the big data framework to create intrusion detection systems using BoT-IoT dataset or other similar datasets.

The following research papers consider many supervised machine learning algorithms to detect attacks in IoT networks.

First, we summarized related work that uses the short version of the BoT-IoT dataset to train an intrusion detection system using machine learning algorithms.

Kumar et al. [5] created a multi-class classification methodology to identify DoS, DDoS, Reconnaissance, Theft attacks, and Normal network traffic. This methodology used feature selection and multi-class classification algorithms. The authors used a hybrid approach for feature selection in which they used Pearson’s Correlation Coefficient, Random Forest mean, and Gain Ratio approach to select features. Then, they joined the results using an AND operation. They applied correntropy to measure the accuracy distinguishing normal and abnormal data samples. Finally, they trained and tested 3 classifiers named Random Forest, XGBoost, and K-nearest neighbors (KNN). This work used the short version of the dataset. The authors highlighted that the approach recognized Theft attacks with 93% of accuracy even though the number of samples of this class was the lowest. XGBoost showed the best results in detecting Reconnaissance, DoS, DDoS attacks with 100% of accuracy.

Shafiq et al. [6] trained and tested five algorithms to detect anomalous behavior using the BoT IoT dataset. This paper is different from others because the authors evaluated how accuracy, precision, TP rate, Recall, and training time were important to select the best algorithm. They used a bijective soft set approach to evaluate the best algorithm considering these five factors. They concluded that the Naïve Bayes algorithm reached 98% accuracy, precision, TP rate, Recall, and the training time was around 4s. The authors used Weka to train and test the classifiers. This paper used the shorter version of the BoT IoT dataset.

Soe et al. [15], in 2020, trained and tested a lightweight model to detect anomalous behavior in IoT devices. This model was designed to run on Raspberry Pi and the authors used the shorter version of BoT-IoT dataset. To train and test the models, the authors created three sub-datasets. Each sub-dataset contained only one kind of attack and normal datasamples. It is necessary to highlight that they considered only DDoS, Theft, and Reconnaissance attacks. Therefore, they created 3 sub-datasets. Then, they extracted the most important features for each subset using correlated-set threshold on *gain-ratio (CST-GR)*. Each subset had different number of features. DDoS features were *drate* and *total number of bytes per destination IP* while Theft features were *state number, total number of packets per protocol, and average rate per protocol per dport*. The authors then trained 4 classifiers named tree-based J48, Hoeffding Tree, logistic model tree, and random forest. The authors reduced the number of datasamples of DDoS and Reconnaissance since it does fit on the Raspberry Pi memory. The authors concluded that the model could detect all kind of attacks with an accuracy of over 99.3% in all the cases. Random Forest was the best algorithm to detect all kind of attacks. This paper has some drawbacks. First, when a new datasample is the input, it should pass through 3 feature extraction and evaluation. This problem increases processing time and unnecessary usage of resources. In addition, the authors down-sampled the number of datasamples for training. For this reason, some important statistical information could be missed.

Bagui and Li [16] developed a framework using Artificial Neural Networks (ANN) and different methods of resampling to obtain models to detect anomalies in IoT networks. Since IoT network datasets are imbalanced, it is difficult to obtain a model which recognizes the minority class with high accuracy. The authors used a Spark cluster and a standalone computer to run their experiments. They used a compact version of the BoT IoT dataset. Their models only tackled a classification problem in which they tried to identify different kinds of attacks and when the traffic was normal. When the experiment was run in a Spark cluster provisioned in AWS, they obtained the best Macro F1-score of around 58% when the sampling method was Random oversampling.

Fatani et al. [17], in 2021, used an innovative feature selection approach called SWARM intelligence with AQUILA optimizer to detect IoT attacks using the BoT-IoT dataset. This methodology is composed of some stages. The first stage is called feature extraction in which they used a Convolutional Neural Network to extract meaningful features from the original raw data. They extracted the features generated from the last fully connected layer which had 64 neurons. Then, they ranked these features to select the most important ones using AQUILA optimization. Finally, they used the final features to train a machine learning algorithm. They used the shorter version of BoT IoT dataset reaching 99% of accuracy in the training and testing dataset. However, if we look at the confusion matrix the accuracy to detect normal class was 60.7% and 85.7% to detect Theft. The overall accuracy was high since the dataset is highly imbalance in nature. However, the model could not generalize well the minority classes.

We can conclude that the authors in references [5,6,15,16,17] can detect accurately the attacks from BoT-IoT dataset. Nonetheless, all of them use the shorter version of the dataset. Thus, we cannot expect that these models will perform well with unseen data.

Next, we summarize some research papers that use sampling methods to reduce the class imbalance of the dataset. In addition, we include some papers that use GAN to generate new datasamples.

Zixu et al. [18], in 2020, developed a novel approach to recognize anomalous behavior locally in each IoT device. They used a GAN to find the best data distribution representation of the data using normal network traffic in each device. The GAN network consisted of a generator and a discriminator. The input to the generator was random data with normal distribution. The authors defined 100 features as an input to the network. The output of the generator corresponds to the number of features which would be the input to the discriminator. Since the authors defined 9 features (flag, state, mean, stddev, max, min, rate, srate, drate), the output of the generator was 9. The generator network was composed of 2 hidden layers with 1024 and 256 neurons on each layer. The discriminator neural network was symmetrical with the generator. After training the discriminator and generator locally, the weights of the generator were sent to the central authority. This entity aggregated the weights of each of the local neural networks. The central authority with a random input generated new samples. These samples were passed through an autoencoder. The main goal of the autoencoder was to learn a representation of the distribution of the data using backpropagation. The autoencoder had two parts - the encoder and the decoder. The encoder part encoded or reduced the size of the original input. On the other hand, the decoder decoded the reduced input to create a vector with the original input. The error was calculated with the predicted decoder output and the original input. Depending on the error, the authors defined a threshold to identify benign and malicious traffic. After creating the autoencoder model, this model was spread among all the nodes. This model was able to discriminate between benign and malicious signals. The authors compared the results with other anomaly detection techniques such as One-class SVM, Isolation Forest, K-means clustering, and Local Outlier Factor. The results showed that the proposed model achieved improved performance when compared to the other models.

Ferrag et al. [19] created a methodology to reduce the impact of imbalanced datasets in IoT networks anomaly detection. They proposed a model which consisted of 3 models. In the beginning, two models ran in parallel. The first model only identified between normal and malicious behavior. The second model labeled all rows of the training dataset as benign or one of the different categories of attacks. The classification outputs of the two models were appended to the dataset as features. Then, the third model was created training the features and the results of the two prior models. The classification algorithms used to build the model were REP tree, JRIP, and Forest PA. The authors trained and tested the model using the shorter version of the BoT-IoT dataset reaching a low false alarm rate and higher detection rate.

Prabakaran et al. [20] proposed a methodology that used GAN to discriminate between normal and malicious IoT traffic. The authors used the shorter version of the BoT IoT dataset in their approach. In the beginning, they labeled all the rows as benign or attacked to create one dataset. Then, they created another dataset labeling all rows as benign or one category attack. Finally, they normalized and joined both datasets. The final dataset was used to train a GAN network. The authors changed the discriminator loss function to reach a good performance of the model. They showed that the accuracy reached by the discriminator was greater than other deep learning models such as Convolutional Neural Network (CNN), autoencoder, KNN, MLP, ANN, and Decision Trees (DT). The accuracy shown in the paper was around 92%.

Ullah and Qusay [21] developed one of the most complete methodologies that used GAN networks for anomaly detection in IoT devices. In their methodology, the authors generated more data samples from the minority class using one class conditional GAN. In addition, they generated normal and anomalous data samples training a conditional binary GAN network. To train the binary GAN network, they reduced the size of abnormal data samples to have a balanced dataset. Finally, they used a multiclass classification GAN network which consisted of multiple binary GAN networks. After generating the new data samples using each GAN network from the three configurations, they trained a feed-forward neural network with a deep architecture.

Although the papers cited in [18,19,20,21] proposed the best methodologies to solve class imbalance problem, the authors did not evaluate their models with the entire dataset.

Finally, we described next some research papers that have used big data frameworks for intrusion detection.

Belouch et al. [22] used Apache Spark to train and test 4 classifiers using the UNSW-NB15 dataset for intrusion detection modeling. They concluded that Random Forest was the most accurate algorithm with 97% of accuracy and 5.69 s training time while Naïve Bayes was the worst algorithm with 74.19% accuracy and 0.18 s training time. The authors used the shorter version of the dataset with 257,340 records.

Haggag et al. [23] proposed the usage of Spark platform for training and testing deep learning models in a distributed way to detect intrusion detection attacks. The authors used the NSL-KDD dataset to train MLP, RRN, and LSTM deep learning models. In addition, the authors added one stage called class imbalance handling using SMOTE. It is necessary to highlight that Spark does not have deep learning capability. Therefore, the authors used Elephas to train and test the deep learning models. To use Elephas, the input should have 3 dimensions. For this reason, RDD form as input to Elephas was the solution. The authors showed that the average F1-score detection was 81.37%.

Morfino et al. [24] proposed an approach to train and test machine learning models in Spark to detect SYN/DOS attacks in IoT networks. They used MLIB to train binary classifiers. The data trained was around 2 million of instances. The authors demonstrated the Random Forest is the algorithm provided the best accuracy of around 100% and the training time was 215s. Our dataset is different since it contains more than 50 million of records.

The following paper is the most relevant work we found in which the researchers used Hadoop–Spark to train and test the entire BoT-IoT dataset.

Abushwereb et al. [25] used MLIB from Spark to train the shorter and larger version of BoT-IoT dataset. The authors proposed a methodology in which they removed duplicated values and rows with missing and unkown values, normalized the data with min-max normalization and applied feature selection using chi-square. The authors then trained machine learning algorithms named RF, DT, and NB using 70% of the data. Finally, they evaluated the accuracy of the algorithm with the 30% of the remaining data. The framework used by the authors was created on Google Cloud platform. The hadoop-spark cluster consisted of eight Vms with an overall Ram of 16Gb. After training and testing the multi-class classification problem, the authors concluded that the overall F1-score was 77% for DT and 73% for RF. The F1-score decreased since normal and theft had much less number of datasamples when compared with other classes. The authors indicated that its model could detect Theft attacks only 23% of F1-score and Normal datasamples with 71.8%. This reference presents a similar approach as the present paper; however, the theft and normal class accuracy are quite low.

## 3. Methodology

This section is divided into two subsections. Section 3.1 explains two methodologies to train and test multi-class classification using multi-class and binary algorithms available in Spark. Section 3.2 explains the data generation of the minority class using CTGAN.

### 3.1. Methodologies to Train and Test Multi-Class Classification Using Multi-Class and Binary Algorithms

The proposed framework is composed of two main systems as we can see in Figure 1, namely anomaly detection and machine learning for intrusion detection.

The first system is used to identify if a datasample is malicious or normal. In this system, we train a one-class SVM using only normal datasamples with specific hyper-parameters. Next, we evaluate the detection accuracy of all the malicious classes in the dataset. If the accuracy is not the value that we expect, we change the hyper-parameters of the one-class SVM again. We do this process in a loop. It is known that hyper-parameters in one-class SVM are important since small changes can extremely modify decision boundaries. The evaluation is done using a distributed framework in Spark. After choosing the model which provides the best accuracy, we use the model to detect if a datasample is normal or malicious. If the datasample is malicious, it goes to the second system, named machine learning for intrusion detection, to be classified as DoS, DDoS, Theft, and Reconnaissance. In the second system, we train a multi-class classification to determine the class of attack that the sample belongs. In the present paper, we do not change the anomaly detection system. We change the flowchart of the machine learning for intrusion which trains and tests multi-class algorithms and multi-class using binary classification algorithms using Hadoop–Spark cluster. In reference [13], we trained and tested a Random Forest which is a multi-class classification algorithm. It is called multi-class because it handles more than two labels. In this work, we expand the research by considering the evaluation of other algorithms that were implemented in Hadoop–Spark. However, some of these algorithms are binary; thus, it is necessary to wrap a binary classifier within One vs. Rest (OVR) to classify multiple classes. We create two methodologies depending on the type of the classifier available in Spark since it contains different steps:Multi-class classification algorithms: Logistic regression, Naive Bayes, Decision Tress, and Random Forest.Binary classification algorithms: Decision Trees, Logistic Regression, Gradient boosted tree, SVM Linear, Naive Bayes, and Random Forest.

Two methodologies are shown in Figure 2 to train and test multi-class classifiers in Hadoop–Spark. The flowchart in Figure 2a shows the feature selection flowchart which is the first stage in Methodology 1 and 2. Feature selection helps to reduce the number of features in the dataset. The BoT-IoT dataset has around 29 features; thus, if we use all these features to train a model, some features can introduce noise in the model. For this reason, as it was shown in reference [13], 8 features were selected. To find the best 8 features, we use the flowchart in Figure 2a. Feature encoding stage is used to transform categorical features to numerical. Vector assembler stage is used to concatenate all the features and labels in a spark format. Scaler stage is used to apply standard scaler to each feature of the dataset. Standard scaler uses the mean and variance to normalize each feature of the dataset. Finally, Random Forest algorithm is used to rank the features in the dataset.

#### 3.1.1. Methodology 1: Multi-Class Classification. Spark Multi-Class Classifiers

The Methodology shown in Figure 2b is used to train and test Spark multi-class classifiers. This flowchart is composed of the following stages namely (i) feature selection, (ii) feature encoding, (iii) vector assembler, (iv) scaler, and (v) multi-class classification. The feature selection stage is useful for feature reduction. All of the steps of this stage were explained in Figure 2a. In this stage, we select the 8 most important features of the 29 features using the Random Forest algorithm. After selecting the 8 most important features, we apply feature encoding to categorical features. In Spark, we use stringindexer to solve this problem. Stringindexer maps each category to numerical values. We did not use one-hot encoding since we can obtain very sparse arrays which cannot be processed for some algorithms. Next, we apply vector assembler to the features. In this stage, we concatenated all 8 features in a single array. This stage is necessary for Spark. Then, we scale the features using standard scaler normalization. This kind of normalization scales each data sample according to the mean and standard deviation of the entire dataset. Finally, we use four multi-class classifiers to train and test our algorithms which are explained mathematically as follows:

**Naïve Bayes Approach:** It uses the Bayes Theorem to compute the conditional probability distribution of each feature given each label. Each data sample is composed of 8 features represented by a vector $x=(x_{1},x_{2},x_{3},x_{4},x_{5},x_{6},x_{7},x_{8})$. We assume that the features are independent among them. Thus, the probability that a class $C_{k}$ happens given the features are given by the following formula [26]:(1)$$p\left(C_{k}|x_{1},x_{2},x_{3},x_{4},x_{5},x_{6},x_{7},x_{8}\right)=\frac{1}{Z}p\left(C_{k}\right)\prod_{i=1}^{n=8}p\left(x_{i}|C_{k}\right)$$
where
(2)$$Z=p\left(x\right)=\sum_{k}p\left(C_{k}\right)p\left(x|C_{k}\right)$$
*k* represents the number of classes that we have in this case 5. Since we use Multinomial naive Bayes, we cannot pass negative values to the algorithm. For this reason, we use min-max normalization. Min-max normalization transforms the features in the range 0 to 1.

**Decision Trees:** It is called a Decision Tree because it has many decision leaves. Each decision leaf is built using a measurement of impurity. When a new data sample enters the system, it goes through each leaf which has a conditional statement. Then the data sample is classified.

**Random Forest:** A Random Forest algorithm is a set of decision trees in which a decision is taken by all of them using a voting schema [27]. Each decision tree is built with a bootstrapped dataset and a subset of the features to build the tree. We can obtain a variety of trees which can take better decisions than a single tree. The more pure features will be at the top of each decision tree. It means that these features are more important to identify between normal and malicious network traces. Therefore, the random forest can rank the importance of the features.

**Logistic Regression:** Multinomial logistic regression finds the correct class from more than two classes given several samples *N* with several features *M*. If we have an input matrix *X*, which is composed of several samples *N* and several features *M*. Then *X* has dimension N×M. The idea is to find the best values of the matrix *W* to obtain the labels that we have as ground truth. In the beginning, the values of *W* are selected randomly. To find the best values of *W*, we have to find a loss function and a gradient. We explain how we can obtain the gradient. First, we calculate the product of *X* and *W*. We denoted the product as Z=(XW). We take the softmax function for each row of the new matrix *Z*. The softmax function gives us the probability of each class given a sample. Thus, the row will sum up 1. As we have known beforehand, our problem is supervised since we have the labels for a given sample. Thus, it is possible to find the likelihood function of *Y* given *X*.
(3)$$p\left(Y_{i}|X_{i},W\right)=P_{i,k=Y_{i}}=softmax\left(X_{i,k=Y_{i}}\right)=\frac{exp(x_{i}*W_{k=Y_{i}})}{\sum_{k=0}^{C}exp\left(x_{i}w_{k}\right)}$$

The formula above was for a single data sample. Considering all the data samples in the dataset, we have the following formula
(4)$$p\left(Y|X,W\right)=\prod_{i=1}^{N}\frac{exp(x_{i}*w_{k=Y_{i}})}{\sum_{k=0}^{C}exp\left(x_{i}*w_{k}\right)}$$

The gradient calculation is as follows. It is necessary to highlight that the gradient $W_{k=Y_{i}}$ is the identity matrix $I_{|Y_{i=k}|}$
(5)$$\nabla W_{k}f\left(W\right)=\frac{1}{N}\sum_{i=1}^{N}\left(X_{i}^{T}I_{|Y_{i}=k|}-X_{i}^{T}\frac{exp(x_{i}*w_{k})}{\sum_{k=0}^{C}exp\left(x_{i}*w_{k}\right)}\right)+2\mu W$$

#### 3.1.2. Methodology 2: Multi-Class Classification Using Binary Classification Algorithms Available in Spark

We divided the stages for multiclass classification using binary classifiers into two pipelines since the wrapper One vs. Rest (OVR) available in Spark only accepts “features” and “label” as column names. Thus, the first pipeline denoted as pre-processing prepares the data. Then, we renamed the features columns as “features” and labels as “label”. Finally, we train and test the algorithms.

#### 3.1.3. Pipeline 1

The first pipeline denoted as pre-processing has the same stages as methodology 1 to pre-process the input. The first stage is features selection in which we reduce the number of features from 29 to 8 features. Then, we have feature encoding. This stage transforms categorical to numerical features. The third stage of pre-processing is vector assembler. In this stage, we concatenate the 8 features into 1 vector. The final stage of pre-processing is the standard scaler in which we normalize all the features with the mean and the standard deviation.

#### 3.1.4. Pipeline 2

In this pipeline, we train and test 6 binary classifiers. In Pyspark, we can wrap binary classifiers into an estimator denoted as One vs. Rest (OVR). Then, we can use these binary classifiers to do multi-class classification. We use the One vs. Rest approach to train and test Support Vector Machine (SVM) Linear, Gradient Boosted tree classifier, Random Forest, Decision Trees, Naïve Bayes, and Logistic Regression. In One vs. Rest, we choose one class of all classes. We label this class as positive while all the rest samples are labeled as negative. For this reason, a model will be created for each class. If we have five classes as in our case, we need to create five models. These five models will take the decision when a new sample is input into the system. When a new input enters the models, our input is evaluated for each classifier. The more confident classifier will be defined as the output label. One vs. Rest estimator has some advantages such as: Parallelism: We can train each classifier in a different node in the Hadoop-spark cluster Model interpretability: We can understand which are the factors that affect each class separately. However, class imbalance is one of the biggest disadvantages of OVR. In this paper, we evaluate how class imbalance impacts the accuracy of the models. In this section, we explain the rest of the classifiers that were not explained in Methodology 1.

**Gradient boosted tree:** A Gradient boosted tree takes a decision based on a consecutive decision trees as week classifiers. In the beginning, a based model is created to find the residuals for each datasample in the dataset. The residual is calculated substracting the original classification label minus the probability to be positive or negative label. We assume that we have binary classifiers. Then, a regression decision tree is built with the feature inputs and the residuals. New residuals are calculated for each datasample. Then, a new decision tree is trained using the features and the new residuals. This process continues depending on the number of decision trees specified by the user. After creating many trees, when a new datasample is the input to be classified, all decision trees take partial decision on the final decision.

**Support vector machine:** This algorithm is different from the rest of the classification algorithms since it tries to maximize the width of the gap between two categories. The hyperplane is known as a threshold which is the main hyperplane that divides both classes. Parallel to the hyperplane exists two additional hyper-planes which define the margin. The main goal of the algorithms is to maximize the distance between both classes minimizing the classification loss. The advantage of this algorithm is that it can handle outliers and admits misclassification. Thus, this algorithm can generalize better unseen data.

### 3.2. Oversampling CTGAN

Class imbalance is one of the major problems in IoT attack detection specifically if we use BoT-IoT dataset. Some authors try to solve this problem using oversampling or undersampling techniques. However, the new data samples generated cannot represent the original distribution. For this reason, we evaluate CTGAN [28] in the present work. It is noteworthy that we train and evaluate the new dataset with the entire dataset of BoT-IoT which is unique in the literature. We use the Methodology 2 shown in Figure 2c to do that.

GAN networks are used for data generation specifically in image analysis. These networks work fine in this field since pixel images values converge in a Gaussian distribution. Thus, normalization and training work fine. In our case, we want to generate new tabular data samples. The term tabular means that we have many features which include categorical and numerical. If we want to use normal GAN networks to generate new data samples of a minority class, we can face some problems.

Since we have different features in the dataset, each feature can follow a different distribution and can be discrete or continuous. Hence, we have a mix of types and distributions in a tabular dataset. Thus, it is really difficult that the GAN network can learn the multivariate distribution to generate new data samples. In addition, each feature can have more data samples of one class than the other class. Thus, we have a class imbalance dataset.

Furthermore, tabular datasets can have multiple continuous data features which do not follow a Gaussian distribution. Thus, if we normalize with min-max or standard scaler datasets, we can lose a lot of information if the feature contains some outliers. Thus, it is really important to select an accurate method for normalization. Since we can have many features, the problem to solve is converted into a high-dimensional problem. Thus, it is really hard that GAN networks can learn these patterns to generate new data samples. To solve all the problems above, Lei Xu et al. [28] proposed a new methodology to generate new data samples of minor represented classes using CTGAN. CTGAN changes continuous variables to discrete variables using mode-specific normalization. In this theory, the probability distribution of the variable is approximated using a variational Gaussian mixture model. After this a GAN network is trained with transformed continuous variables and discrete variables one-hot encoded. In this research, we compare CTGAN with other oversampling methods such as ADASYN and SMOTE. In the following subsections, we explain both oversampling methods.

#### 3.2.1. Synthetic Minority Over-Sampling (SMOTE)

This oversampling method generates new data samples interpolating values inside a K neighbors cluster. In the beginning, SMOTE finds a ratio of the minority class. Then, it selects all data samples belonging to the minority classes. It is defined as a K which is the number of neighbors from each data point of the minority class. In each group of K, the data points labeled with the minority class is chosen. Then, the algorithm finds a linear distance between the chosen minority data point and the other minority data points inside the cluster. Next, the algorithm chooses data points on the linear lines for each minority datapoint. The equation used to generate new datasamples is $s_{i}=x_{i}+\left(x_{zi}-x_{i}\right)\delta$. Where $x_{i}$ is a sample of the minority class selected from K neighbors. $X_{zi}$ is another data point of the minority class selected randomly in the K neighbors. δ is a random value selected that can take values between 0 and 1. SMOTE suffers from some disadvantages such as only 1 data sample of the minority class in a neighborhood. One solution to solve this problem is selecting more K neighbors. In addition, we may have outliers in the dataset; thus, if we generate more data samples from outliers, the generated data will be outliers [29].

#### 3.2.2. Oversample Using Adaptive Synthetic (ADASYN)

This algorithm is similar to SMOTE. However, it generates more data points when minority class data points are the majority in a cluster of k neighbors. ADASYN is also sensitive to outliers as SMOTE. ADASYN solves the problem of only 1 data sample in a neighborhood since this algorithm analyses where we have more data samples to generate new data [29].

## 4. Experiments and Results

We divided this section in three subsections. The first subsection describes the BoT-IoT dataset and contributions of our previous work published [13], second subsection explains the results obtained when we use multi-class and binary classification algorithms in Spark. Finally, the last subsection explains the results when we use oversampling of the minority class using CTGAN and data augmentation techniques.

### 4.1. BoT-IoT Dataset and Previous Contributions

We use the large version of the BoT-IoT dataset to train and test the anomaly detection and the machine learning for intrusion detection systems. We can see the data class distribution in Figure 3.

This dataset is highly imbalanced since normal and Theft are minority with a proportion of 1/4037 and 1/24280, respectively, if we compare with DDoS number of datasamples. In addition, the original dataset contains 32 network traffic features. In our previous work [13], we applied feature selection with Random Forest reducing the number of feature to eight named *state, proto, bytes, dport, sbytes, dur, sum,* and *max*. We concluded that eight features are sufficient to avoid noise, reduce the time of training, and keep an accuracy over 90%. As we described in Figure 1, first we implemented anomaly detection using one-class svm in Hadoop–Spark to identify normal from malicious datasamples. We can conclude that we can detect Normal and Theft attacks with 98.31% and 96.85% of accuracy, respectively; although, we only selected two features for the training.

### 4.2. Train and Test Multi-Class Classification Using Multi-Class and Binary Algorithms Available

Before training binary and multi-class classification algorithms, we split the data of the entire dataset in 70% for training and 30% for testing with stratification. It means that we select the same proportion of datasamples depending on the number of datasamples of each class. The metric used to measure the accuracy of all the approaches is the F1-score since this metric takes into consideration the precision and recall in other words the false alarm rates. In addition, this metric is the best to evaluate imbalanced datasets. In addition, for the purpose of comparison with other related work we chose this metric. As we described in the methodology section, Spark has binary classification algorithms with OVR and multi-class algorithms to train and test multi-class classification datasets. It is possible to have the same algorithm available as multi-class or binary with OVR. To illustrate, we can train and test a random forest using the multi-class algorithm or a binary random forest wrapped with OVR in Spark. However, the time of training is different. For this reason, we consider two evaluation parameters the training time and the accuracy. Table 1 shows the value of hyperparameters for each algorithm that were used in our experiments.

As we can see in Figure 4, Random Forest algorithm performs with 93.55% of accuracy which is 5% greater than the Gradient Boosted tree. In addition, Decision Trees results are at least 10% less than Random Forest. If we compare Naive Bayes and Logistic Regression results, Random Forest outperforms them with around 40% of accuracy.

After comparing the accuracy, we conclude that Random Forest algorithm using multi-class and binary with OVR classification provides us with the best accuracy. However, as we can see in Figure 5, the training time for Random Forest if we use binary classifier with OVR is five times more. Thus, we conclude that multi-class Random Forest algorithm provides the best accuracy and acceptable training time. Since the BoT-IoT dataset is extremely imbalanced, the overall accuracy although is high, the accuracy to identify minority classes is low. In the results found, the F1-score accuracy to detect Theft attacks is 41.83%. We can conclude that the number of datasamples for theft attacks is not enough to create a robust model. This problem is solved in the next section in which we use CTGAN to generate more datasamples of minority classes.

### 4.3. Oversampling CTGAN

As we described in the previous subsection, Random Forest using a multi-class algorithm in Spark outperforms other algorithms. However, the F1-score accuracy to identify theft attacks is 41%. In this section, we created more datasamples for minority classes. We designed the following two experiments. In the first experiment, we generate 10,000 additional data samples from the normal dataset and 17,000 additional data samples from the Theft dataset using CTGAN. Then, we joined the new samples with the original data samples and we trained and tested a random forest multi-class classification. In the second experiment, we generated theft data so that the number of theft data samples were the same as the original number of normal data samples.

As we can see in Figure 6, although we generated more datasamples (normal and theft) in the first experiment, the F1-score accuracy was not as good as than when we balance the theft datasamples without augmenting normal datasamples. The best obtained F1-score overpasses with around 50% to detect theft attacks if we compare when we did not generate new datasamples with CTGAN.

Finally, we compared the CTGAN results with ADASYN, and SMOTE oversampling methods. We generated the same number of minority datasamples as in experiment 2. The results of the comparison are shown in Figure 7. We can conclude that when we generate more data samples with CTGAN, the theft detection increases by around 50%. We can see that ADASYN is better than SMOTE to generate new datasamples. However, the CTGAN generates better datasamples than that of ADASYN and thereby provides the best classification results.

We compare the best model obtained in the present paper with other related work in Table 2. As we can see references [5,6,15,17] use the shorter version of the dataset; thus, the F1-score to detect DoS and DDoS outperforms our approach because the quantity of datasamples is much less; thus, the classifiers can determine better boundaries. However, our methodology can detect Theft attacks better than all the other related work as shown in Table 2. In addition, our approach can detect Normal datasamples better than other approaches except [5] with only 2% difference. As we described before that comparing the accuracy of our approach with papers that use the shorter version of the BoT-IoT dataset is not fair since the shorter version does not contain enough statistic characteristics that represent the population.

We can make a fair comparison only with reference [25] since the authors of this paper use the entire BoT-IoT dataset and Hadoop–Spark framework to process the data. We can see that our approach outperforms F1-score accuracy to detect Normal and Theft attacks with significant differences. Our approach detects Normal datasamples with around 26.2% F1-score more than [25]. We need to highlight that our approach is 4 times more accurate to detect Theft attacks than reference [25] since we use CTGAN to generate real datasamples for augmentation. Our approach is not so good in detecting DDoS and DoS attack samples; nonetheless, the difference is not significant. The overall F1-score accuracy of our work is 96.77% while for reference [25], it is 76.57%. If we compare the overall accuracy of our work with other papers except reference [15] since this work does not evaluate Normal and DoS, only reference [5] surpasses our approach by 1.8%; however, it uses the short version of the dataset. We cannot compare our approach in terms of computational efficiency with [25] since the authors used a pre-configured model in Google cloud that had eight VMs. In our case, we implemented the hadoop spark cluster in our lab with only has 3 workers with 2 cores and 8GB of RAM.

Finally, we ran the experiment five times with different seed to select the training and testing datasets using stratification. The results are shown in Figure 8. As we can see, the results do not change much if we compare them with Table 2 considering the size of this dataset this was expected.

## 5. Conclusions and Future Work

In this paper, we propose a combination of systems and strategies to reach the best accuracy for detecting different attacks from the BoT-IoT dataset with an average F1-score accuracy of 96.77%. Our work differs to other related papers since our work trains and test a model with the entire BoT-IoT dataset. We use a Hadoop–Spark strategy to reach this goal. Training a model with more data means robustness to identify new unseen data.

We show that Random Forest is the best algorithm to train extremely imbalanced datasets. In addition, we demonstrate that OVR wrapper is useful to train multi-class problems with binary classification algorithms. However, training time with OVR (binary classifier) was five times more than that of multi-class classification. For this reason, we select Random Forest algorithm from Spark multi-class algorithms to train the model. Although the accuracy is high, if we analyze each class, we notice that minority classes are affected. For this reason, we propose the usage of CTGAN to generate new datasamples. CTGAN is different to other oversampling methods since it creates datasamples from the original distribution. We compare CTGAN with other oversampling methods such as SMOTE and ADASYN concluding that CTGAN is more accurate to generate datasamples. With CTGAN, Random Forest provides an F1-score of 96.77%. Finally, we compare the results of our approach with other recent related work concluding that our approach is more accurate in detecting minority class, e.g., Theft and Normal classes. It is necessary to highlight that our approach uses the entire dataset; thus, our methodologies can be more robust to unseen data.

Since our research was focused on attack detection using machine learning, in the future, we want to solve this problem using deep learning models which are not available in MLIB spark library. For this reason, our plan is to develop a blockchain-based federated learning approach to train our models with high accuracy, security and less processing time.

## Figures and Tables

**Figure 1 sensors-22-07726-f001:**
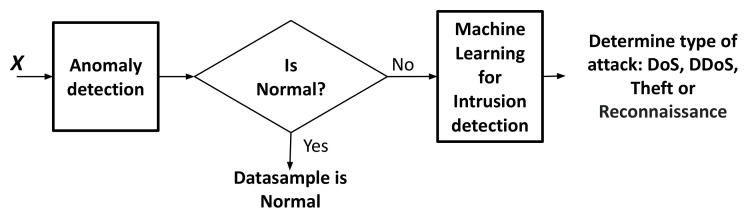
General framework.

**Figure 2 sensors-22-07726-f002:**
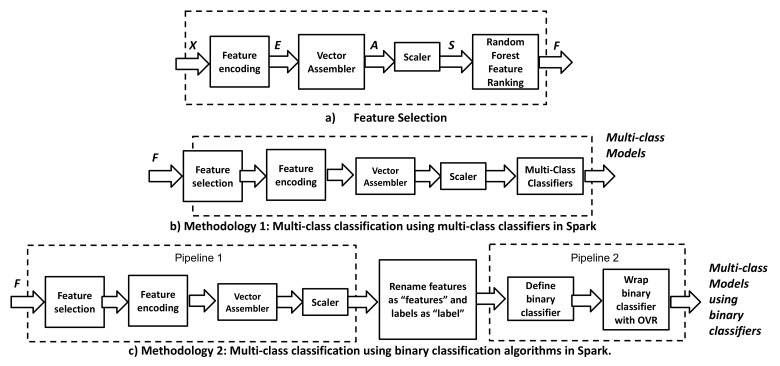
Intrusion detection using machine learning methodologies.

**Figure 3 sensors-22-07726-f003:**
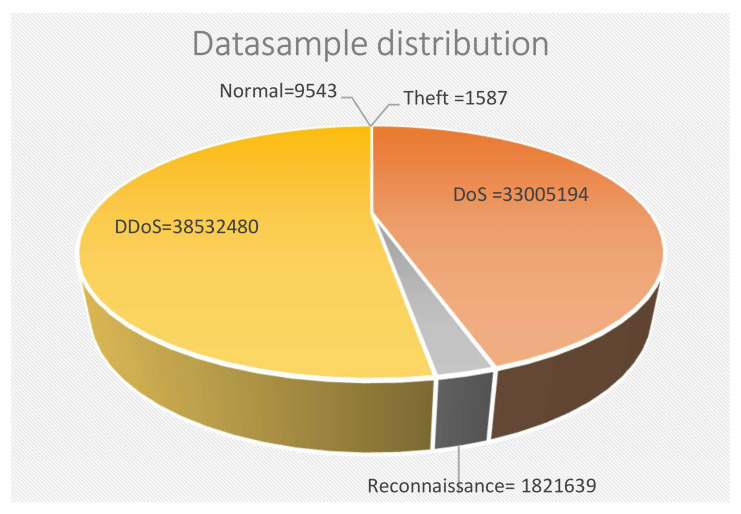
Datasample class distribution.

**Figure 4 sensors-22-07726-f004:**
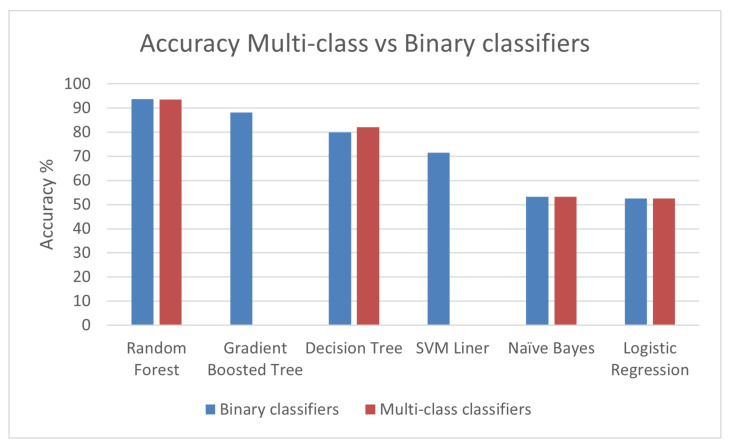
Accuracy for Multi-class vs. Binary classifiers with OVR in Spark.

**Figure 5 sensors-22-07726-f005:**
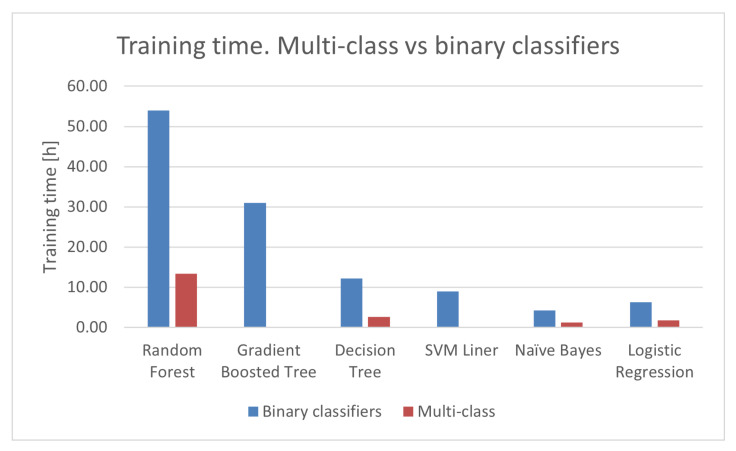
Training time for Multi-class vs. Binary classifiers with OVR in Spark.

**Figure 6 sensors-22-07726-f006:**
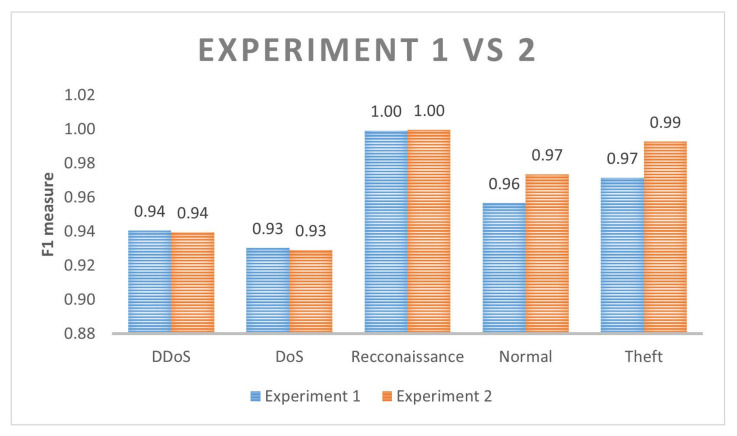
Random forest results after CTGAN oversampling.

**Figure 7 sensors-22-07726-f007:**
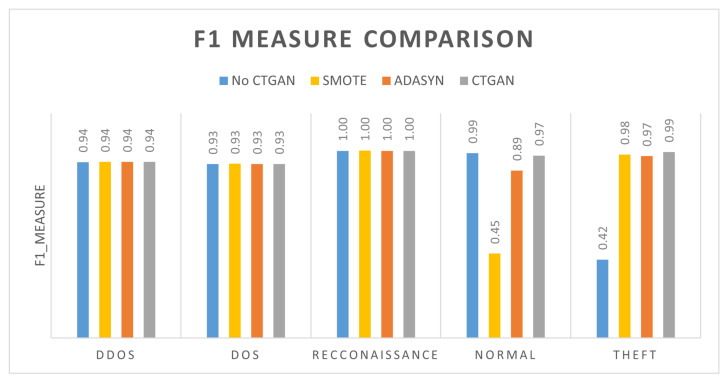
F1 measure after Oversampling. Comparison among SMOTE, ADASYN, and CTGAN.

**Figure 8 sensors-22-07726-f008:**
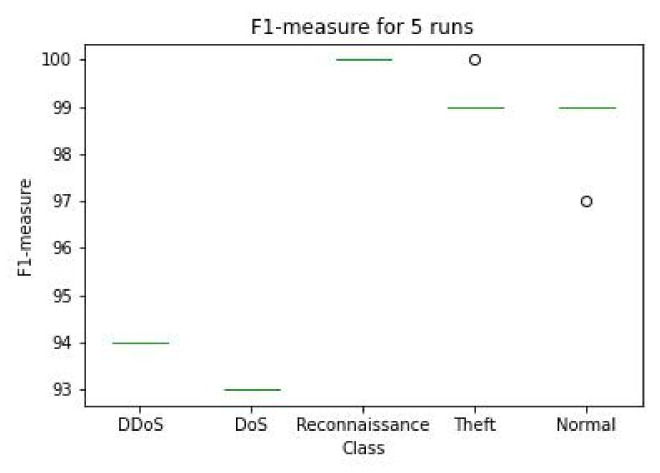
F1 measure after running the five different experiment with different seeds.

**Table 1 sensors-22-07726-t001:** Algorithms Hyper-Parameters.

Algorithm	Hyper-Parameters
Random Forest	numTrees = 30, maxDepth = 30, Impurity = Gini
Decision Tree	maxDepth = 5, Impurity = Gini
Gradient Boosted Tree	maxDepth = 5, Learning_rate = 0.1, Impurity = variance
SVM Linear	regParam = 0.1, kernel = Linear, HingeLoss
Naive Bayes	N/A
Logistic Regression	elasticNetParam = 0.8, penalty = Elasticnet

**Table 2 sensors-22-07726-t002:** Comparison of detection rate with other related work.

Ref	Normal [%]	DDoS [%]	DoS [%]	Reconnaissance [%]	Theft [%]	Algorithm	Dataset
Shafiq et al. [6]	75	98	100	81	93	NB	Short version BoT-IoT
Soe et al. [15]	-	99.9	-	99.9	98.18	Random Forest	Short version BoT-IoT
Kumar et al. [5]	100	100	100	100	93	XGBoost	Short version BoT-IoT
Fatani et al. [17]	60.7	99	99	99	85.7	Aquila optimizer (AQU)	Short version BoT-IoT
Abushwereb et al. [25]	71.8	99.9	99.13	88.83	23.2	MLIB(RF)	Large version BoT-IoT
Our approach	98	94	93	99.86	99	Random Forest	Entire BoT-IoT dataset

## Data Availability

We use for analysis the BoT IoT dataset available in https://cloudstor.aarnet.edu.au/plus/s/umT99TnxvbpkkoE. The data generated after analysis was not published publicly.

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
