# Peer review of "Towards Developing a Robust Intrusion Detection Model Using Hadoop–Spark and Data Augmentation for IoT Networks†"

_sensors, 2022, doi:10.3390/s22207726_

Round 1

Reviewer 1 Report

The authors explain their proposed framework for the introduction detection model with machine learning classifiers and data balancing method. This work is interesting; however, the following flaws have to be addressed.

- In section 1, the authors mentioned that "the processing of data samples in the selected cannot possible without big data tools.".  Do you have any references for that? Some works have already used that dataset without applying any big data tools. It would be better if you have some other reasons rather than using the word, “impossible”, to point out why you believe in using Hadoop-Spark.

- In section 3, you can describe your selected dataset. I think that the nominal features are included in it. According to Figure 1, the first process is for anomaly detection with one-class SVM. In this regard, how does it work with nominal attributes?

- In the cases of applying machine learning algorithms, some more description is necessary, like which method did you choose in the Decision Tree, how many trees did you set up in Random Forest, and so on. The different configurations of algorithms will be consequent on your results.

- In section 4, it would be better if you compare the original distribution and updated distribution of the dataset with a table or a graph.

- Figure 5 has to be revised, what are experiments 1 and experiment 2?

- Please be brief and describes the previous extension of your published paper in the manuscript, and follow the journal guideline. E.g., you need to describe the feature selection process, which features did you select? 

- In addition, in the experiment results. how to evaluate your model? how to distinguish the train and test data?

Author Response

Dear,

We are submitting our revised manuscript entitled “Towards developing a robust intrusion detection model using Hadoop-Spark and data augmentation for IoT networks.” to the Journal MDPI on Special Issue "Communication, Security, and Privacy in IoT" for review and possible publication. In this document, we explain how we have taken the reviewers’ feedback into account in the revised paper.

We thank all the reviewers for their efforts and the useful comments. We carefully addressed all the comments to improve the technical quality and general readability of the paper.

Reviewer 1

Comment 1: In section 1, the authors mentioned that "the processing of data samples in the selected cannot possible without big data tools.".  Do you have any references for that? Some works have already used that dataset without applying any big data tools. It would be better if you have some other reasons rather than using the word, “impossible”, to point out why you believe in using Hadoop-Spark.

We justified the usage of Hadoop-Spark with some statistics that we obtained during our experiments. In addition, we modified the context that included the word impossible. Finally, we added a reference which compares Spark performance. Page 2, Section 1. Paragraph 1.

Comment 2: In section 3, you can describe your selected dataset. I think that the nominal features are included in it. According to Figure 1, the first process is for anomaly detection with one-class SVM. In this regard, how does it work with nominal attributes?

We included a brief description of the BoT-IoT dataset. In addition, we included the method of feature selection and the results for the first part of one-class SVM. Page 11, Section 4.1.

Comment 3: In the cases of applying machine learning algorithms, some more description is necessary, like which method did you choose in the Decision Tree, how many trees did you set up in Random Forest, and so on. The different configurations of algorithms will be consequent on your results.

We included Table 1 with the configuration for each algorithm that we use. Page 12, Section 4.2.

Comment 4: In section 4, it would be better if you compare the original distribution and updated distribution of the dataset with a table or a graph.

We included a pie chart in Figure 3 with the original distribution of the dataset. Page 11.

Comment 5:  Figure 5 has to be revised, what are experiments 1 and experiment 2?

Note: It was changed the number of the Figure to 6.

In Figure 6, we compare two experiments. The description of the experiments is in Page 13, Section 4.3. In this paragraph, we explain both experiments. In experiment 1, we generated Theft and Normal additional datasamples while in experiment 2, we only generated theft datasamples for data augmentation.

Comment 6: Please be brief and describes the previous extension of your published paper in the manuscript, and follow the journal guideline. E.g., you need to describe the feature selection process, which features did you select? 

We describe our previous paper on the introduction section Page 2, paragraph 5. In addition, we included a description of feature selection and the dataset in page 11, section 4.1

Comment 7: In addition, in the experiment results. how to evaluate your model? how to distinguish the train and test data?

We included training and testing split in Page 11, Section 4.2. In addition, we included the results for additional runs with different seeds in Figure 8. We included the explanation of these runs in Page 15, Paragraph 2.

Please find attached the latest version of the paper.

Reviewer 2 Report

1.The contribution of the manuscript could be added in the introduction.

2.Some paragraph(line 85-line123)could be improved, where the cited references could be more relevant to the research.(Too few references contain both BoT-IoT datasets and Hadoop-Spark).

3.Insufficient summary and analysis of relevant  papers.Some paragraph(line 85-line123,line 130-line 182)could explicitly point out progress and deficiency  of other works.

4.The evaluation parameter F1 used in the experiment may be not perfect. (only the accuracy rate and recall rate are considered), and some other evaluation parameters like cross validation, scale of training set and testing set could be considered.

5.There is no future work in the end.

6.The chart form is too simple(only histogram).

Author Response

September 27, 2022

We are submitting our revised manuscript entitled “Towards developing a robust intrusion detection model using Hadoop-Spark and data augmentation for IoT networks.” to the Journal MDPI on Special Issue "Communication, Security, and Privacy in IoT" for review and possible publication. In this document, we explain how we have taken the reviewers’ feedback into account in the revised paper.

We thank all the reviewers for their efforts and the useful comments. We carefully addressed all the comments to improve the technical quality and general readability of the paper.

Reviewer 2

Comment 1: The contribution of the manuscript could be added in the introduction.

The contributions of the paper were added as bullets in the introduction section, Page 2 and Page 3.

Comment 2: Some paragraph (line 85-line123) could be improved, where the cited references could be more relevant to the research. (Too few references contain both BoT-IoT datasets and Hadoop-Spark).

We extended the related work section including papers that use big data frameworks in intrusion detection. Page 5, Paragraphs 6,7,8,9. Page 6, Paragraphs 1-3.

Comment 3: Insufficient summary and analysis of relevant papers. Some paragraph line 85-line123,line 130-line 182) could explicitly point out progress and deficiency  of other works.

We included more relevant papers in the area. First, we included papers that use big data tools to create intrusion detection systems with different datasets.

Page 4, Paragraph 3. Page 5, Paragraphs 6,7,8,9. Page 6, Paragraphs 1-3.

Comment 4: The evaluation parameter F1 used in the experiment may be not perfect. (only the accuracy rate and recall rate are considered), and some other evaluation parameters like cross validation, scale of training set and testing set could be considered.

We explain why F1-score is the most important measure in imbalance datasets since it evaluates not only the accuracy but also takes into account recall and precision. In addition, we have used it for the purpose of comparison with other related work. Page 11, Section 4.2.

Comment 5: There is no future work in the end.

We included the future work in section conclusion and future work in Page 15, Paragraph 3.

Comment 6: The chart form is too simple (only histogram).

We included pie chart in Figure 3 and box plot chart in Figure 8.

Please find attached the last version of the paper with the changes

Round 2

Reviewer 1 Report

Thanks for the well-revised manuscript based on the reviewers’ comments.

In the current version, one comment is that:

You can revise the sentence in the abstract, "Since not much related work that combines IoT security, Machine Learning, and Big data is not available ……".

It would be better if you mention some other facts, like, what would be the advantage of these techniques’ combination.

Author Response

We are submitting our revised manuscript entitled “Towards developing a robust intrusion detection model using Hadoop-Spark and data augmentation for IoT networks.” to the Journal MDPI on Special Issue "Communication, Security, and Privacy in IoT" for review and possible publication. In this document, we explain how we have taken the reviewers’ feedback into account in the revised paper.

We thank all the reviewers for their efforts and the useful comments. We carefully addressed all the comments to improve the technical quality and general readability of the paper.

Reviewer 2

Comment 1:  You can revise the sentence in the abstract, "Since not much related work that combines IoT security, Machine Learning, and Big data is not available ……". It would be better if you mention some other facts, like, what would be the advantage of these techniques’ combination.

We explained the contributions in the abstract as it was suggested.
